# Service Provider and "No Accident": A Study of Teachers' Discipline Risk from the Perspective of Risk Society

Penghui Hu [1], Shasha Du [2,*] and Guoxiu Tian [3]

1    School of Sociology, Huazhong University of Science and Technology, Wuhan 430074, China
2    Department of Social and Cultural Studies, Party School of C.P.C. Jiangsu Committee, Nanjing 210009, China
3    College of Teacher Education, Capital Normal University, Beijing 100048, China
*    Correspondence: hankdu123@163.com

**Abstract:** Teachers face a high degree of risk when disciplining students in contemporary China. Under the guidance of risk society theory, based on a qualitative study of teachers at a county town high school in Southwest China, this paper finds that, in the context of shifting responsibility for education from family to school and inequal risk distribution system in school, teachers become a primary risk taker. The culture of the teacher as a service provider with unlimited responsibilities and the institution of "No Accident" in daily management supported by schools and local government is constructing the sense of risk in teachers. The consequences of risky events are unbearable for teachers in most cases, so they have to adopt limited discipline strategies with a focus on risk avoidance. Reconceptualizing cooperative family–school relations and constructing a reasonable risk allocation mechanism in school would be the keys to eliminating teachers' conception of discipline risk.

**Keywords:** risk society; sense of discipline risk; risk distribution; service provider; No Accident

## 1. Introduction

The enormous work-related stress faced by teachers in our day is a worldwide problem. A survey titled TALIS 2018 (Teaching and Learning International Survey) conducted by OECD (Organization for Economic Cooperation and Development) revealed that 48.7% of the 260,000 teachers in 48 OECD countries felt "quite a lot" or "a lot" of pressure in their work. A vital source of such stress in teachers' work is the discipline of students, including educating, managing, and communicating with them, as well as the necessary penalties in teachers' daily routines. The TALIS2018 results show that "responsibility for student achievements" and "maintaining rules on classes" are important stressors for teachers, both of which involve disciplining students [1]. In China, in particular, there have been numerous incidents of student–teacher conflict and punishment for maintaining discipline, with the result that teachers experience high levels of stress in the process of disciplining students, mainly due to the high risk of uncertainty about the consequences of discipline. "Teaching is a high-risk occupation" has been a consensus held by teachers in current China (See relative internet news. https://www.sohu.com/a/467207640_120418387, accessed on 19 May 2021).

Teachers' strong sense of risk has negative effects on their physical and mental health, career development, and the growth of their students. On the one hand, a teacher's feeling of risk can lead to feelings of relative deprivation, injustice, and powerlessness, which in turn can lead to withdrawal from work, and on the other hand, a teacher's sense of risk will negatively affect student well-being and academic performance. It is, therefore, an urgent task for educational researchers to focus on teachers' feelings of risk, to diagnose this social problem, and to formulate appropriate policies to promote healthy and sustainable development of school education.

However, there is currently a lack of scholarly discussion of teacher risk feelings, particularly from a teacher's perspective, which would show how they perceive their

feelings of risk and its causes. It is notable that teachers' sense of risk is closely linked to the intrusion of a "risk society" into education [2]. Therefore, from the perspective of risk society, through mixed research of teachers in a county high school in southwest China, this paper will focus on a teacher's feeling of discipline risk, and three specific questions are examined: (a) how do teachers understand the discourse of "teaching is a high-risk occupation"; (b) how is the sense of discipline risk constructed; (c) and the consequence of teachers' sense of the high-risk job.

## 2. Literature Review

### 2.1. Risk Society: The Approaches of Institutionalism and Culturism

Ulrich Beck, a German sociologist, published his book *Risk Society* in 1986, and he proposed the concept of "risk society", which refers to a stage of modernity that is threatened by the industrialization of global society, even beginning to dominate the main social facts. The concept of risk society sheds light on the relationship between modern industrial society and its own problems that go beyond the scope of social understanding of security. In a risk society, we must take the mitigation and distribution of injuries as the core issue [3,4].

Giddens's explanation of risk society emphasizes that a long-term maturity and overdevelopment of modern social systems have resulted in an artificial uncertainty of risk society via human intervention in various social facts or natural environments. For example, the national state system and totalitarianism, the capitalist economy and economic crisis, military order and nuclear war, and so on. As a consequence, the mechanisms of sociogenesis are highly unpredictable and continue to deepen in various public sectors, such as finance, international relations, epidemics, and also education [5]. Moreover, reflexivity about modern society and its risks become an integral part of Giddens's theoretical system.

The concept of risk, in Beck and Giddens' understanding, is a collection of social establishments, systems, structures, or perhaps a social stage. Examples include the division between early modernity and late modernity [5,6]. They argue that, as influences of human decisions and actions on human society itself, the structure of the main risk evolves from natural risks to human uncertainty [7]. As a result, humans have developed a range of institutions with complicated systems and structures to deal with these risks, named "institutionalization of risk". Furthermore, they conclude that new risks are being produced in these institutions of human coping with uncertainty, as they called "institutionalized risks" [8]. Nevertheless, more to the point, the creation of a wide range of modern institutions or bureaucracies in recent times has produced a suitable ground and normative framework for these two kinds of risks, which have essentially contradictory orientations.

Not only are institutionalists such as Beck and Giddens concerned about risk society [5], but social scientists of culturism also provide some vital explanations. They assume risk society is a cultural phenomenon. Risk culture depends on non-institutional and anti-institutional social states. Its dissemination does not rely on procedural rules and norms but on its substantive value. In risk culture, the governance of social members does not rely on laws and regulations but on some symbolic ideas and beliefs, as social members in the risk culture prefer chaos and disorder in the sense of inequality rather than strict hierarchy and order. Uncertain quasi-social members in the risk culture may be a fragmented collection, and they are not very concerned about their own interests; instead, they merely have fantasies and expectations for a better life in this risk culture atmosphere [9,10].

As in the above studies, even though risk culture is considered to be worth embracing, institutionalist and culturalist scholars have launched a number of critical discussions on the educational occupation and risk society [8,11]. Institutionalists preferentially explain the relationship between occupation and risk society, especially in the education sector, in terms of state regimes, power patterns, and capital control. This is because one of the key purposes of education is the production of human capital, which is at stake for the security of a state's power. Kivinen and Ahola analyze the gap between the ideology of human capital in the Nordic countries and the everyday reality of human risk capital faced by graduates, confirming that despite popular market rhetoric, the national government

and its power supporters remain firmly in control of teachers, higher education, and the labor market [12]. It is thus clear that the education sector is tied to the uncertainties of a country full of politically constructed risks.

Scholars such as Tansey and Rayner, who hold a cultural sight, assume risk as a cultural chaos or disorder that presents a horizontally distributed state of unstructuredness [13]. It is transmitted not through a confirmed order or institution but through a scattering of symbolic meanings and beliefs [14]. Social members immersed in this culture prefer to embrace and practice this decentralized disorder rather than obey a defined system [15]. Fenwick observes the emergence of specific norms of technological internalization and self-regulation that "good" teachers deploy themselves through such programs organized by local teachers' associations. These norms are embedded in the values of individualism, technocracy, reflexivity, and the "aggressive self" that support the neoliberal ideology of contemporary risk society. These trends and cultural values encourage local teachers to internalize goal-oriented practice, reflection, and self-regulation [2]. This cultural ethos, in turn, is mainly determined by popular opinion in the local community and public expectations of a profession or occupation.

As can be seen, risk society is a concept constructed in ways that come from vital social organizers of power, as well as from public opinion and social ethos both institutionally and culturally in the educational field, especially schools and teachers, as they are inextricably connected with fundamental values of a regime and general quality of its citizens.

### 2.2. The Occupational Risks of Teachers

The occupational risk for teachers initially comes from the promotion or implementation of authority policies or regulations, as it may upset the balance between teachers, school administrators, and students. Furnes and his colleagues point out that national authorities promote certain educational policies between teacher education institutions and schools for several purposes: to reduce the gap between theory and practice, to gain new insights, and to develop their pedagogical practices to encounter unforeseen future challenges. They argue that partnerships under such terms might lead to tension and risk, which may widen the gap between stakeholders rather than decrease it. Furthermore, there is a risk of missing opportunities to gain new insights and develop pedagogical practices to encounter unforeseen future challenges [16].

At the same time, a particular cultural discourse and its altering would also weaken the teachers' control over the curriculum and knowledge itself, thereby increasing their professional risk. For example, Mamolo and Pinto provide a discussion of how neoliberal discourse affects teaching and interpersonal risk in schools. This powerful cultural script leaves teachers feeling that they must hold control of all the students, authority, and knowledge of examinations [17]. Evaluation systems and expectations of local teachers by schools and local governments have also become another major risk in their careers. A report by White explores the positive relationship with teachers' performance risks, pressures, and the use of multi-rater evaluation systems in 16 districts ranging from New York City to North Caroline, USA, with widely varying student populations, resources, and policy priorities [18].

For China and other Asian countries, a Confucian cultural cluster, the rate of entering a university is another major pressure on the group of middle school teachers because it is directly related to the status, achievements, or even livelihood of a teacher [19–22]. According to Zhang and Liu, severe test pressure will lead high school teachers to struggle and lose their purpose. The two contradictory but coexisting forms of managerial professionalism strongly influence teachers: examination-oriented professionalism and quality-oriented professionalism [23].

### 2.3. The Risks of Managing and Educating Students by Teachers

Current studies mainly focus on suffering from the perspective of students in schools. Examples include student self-esteem damage, emotional exhaustion, impairment of

healthy personality, physical punishment, suicidal behavior, and so on [24–26]. However, few studies have addressed the damage that teachers suffer when they manage and educate their students. In fact, disciplining students is also a factor in structuring teachers' occupational risk. Physical harm becomes a direct cause in teachers' management and education of students, especially for delinquent students [27]. One study found that teachers were most likely to receive beatings and verbal threats from such students when managing them, especially teachers with administrative positions in China [28]. In addition to this, psychological damage from parents and public opinion is frequently visible [29]. Parents will be more inclined to believe that teachers should take more responsibility in a conflict or even that there are serious ethical problems with teachers acting out of a self-defense mindset in disciplinary actions. This attitude, full of subjective value judgments, is more likely to expose teachers to greater public pressure and emotional crisis. Finally, the administrative regulations within the education system and the pressure from their superiors have brought resistance to teachers' discipline of students [30,31]. As a part of the education system, strict codes of conduct and assessment standards have brought shackles for teachers to discipline students. At the same time, school leaders and superiors, out of consideration of their own group interests, are also putting some pressure and burden on the interaction between teachers and students.

*2.4. Research Gap in Previous Studies*

From these studies, it is not difficult for us to find that cultural and institutional issues are closely related to the sense of risk that teachers have in disciplining their students, such as the school evaluation systems and the professional expectations of parents in a specific culture. However, there are still numerous ambiguities about the risks of discipline and its mechanisms for teachers in China that need urgent clarification. It is worth exploring further what aspects of discipline risk are embodied in teachers and how it is constructed. In particular, two aspects are worth studying. One is the bureaucratic status presenting as a public educational institution, such as a county high school in China, which is an administrative part of the local government. The other is the secularization of the professional role of teachers in a time of great cultural change in China, with a rapid shift in parents' expectations of their professional role.

Undoubtedly, the study of the sense of risk among high school teachers is a pressing academic task. This paper aims to explore the institutional structure, cultural context, and hidden mutual mechanism behind the experience of county town teachers' sense of risk in disciplining students and to examine their discipline strategies.

**3. Methodology**

*3.1. Research Site*

This study is based on a three-year research project (2017–2019) on teachers' emotional labor in L High School, a public high school in county town of L County. L County, in southwest China, has been in a state of mild poverty, which has led to a large exodus of labor. Correspondingly, the county is full of elders and children left behind, and as a result, most of the students at L High School live with their grandparents and other relatives.

Our chosen high school shares a common reality with other high schools, which are located in county town, with inadequate financial support in mainland China. In other words, these kinds of counties face similar structural situations, specifically, such as a poor level of economic development, a large labor exodus with a population left behind, a huge rural population moving to county towns or cities, and a common educational reality that their educational standards differ greatly from those of developed cities, which is a principle cause for the loss of qualified teachers and smart students leading to a further hindrance of local education's prosperity.

### 3.2. Theoretial Frame

The dual institutional and cultural perspectives of risk society theory provide the following theoretical guidance for our study.

Firstly, the sense of risk in modern society is a consequence of a unitary organization, system, institution, regulation, and their practices of rewards on social members, as well as punishments. It is indubitably an empirical setting, and such institutions and their practices are based on several fundamental prescriptions of a regime for the power and status imposed on its members. The perception of risk in the teaching field is a result of the regulation and operation of the Chinese government and its auxiliary public system as the power and status of the teaching community, in a great measure derived from the empowerment of the government. In our case, the factors that shape a teacher's perceptions of discipline risk include official rules and their invisible practices. The former includes laws and regulations that govern the teaching profession, the rules for teacher promotion and evaluation, and the practices of school leadership intertwined with other power holders.

Secondly, modern social risk, as a legitimate cultural phenomenon, provides its embedded members with unique role expectations, cultural rules, paths of changing values, and relative legitimacy. Risk culture offers us an objective and empathetic interpretive view of contemporary chaos as constructivist in its existential legitimacy for the actors involved. For instance, the cultural expectation of teachers has undergone a dramatic modern alternation from traditional Chinese spirit as the marketization and secularization of education did lead to the birth of modern education in China. However, it has led to a culture of risk between teachers and parents. Furthermore, it features chaos and ambivalence, creating a set of discourses and action strategies that favor the parents in primary and secondary education. Parents recognize, approve and use this risk culture to create chaos in order to claim certain rights for themselves and their children.

Thirdly, to better understand modern social risk, institutional and cultural perspectives can provide us with a more thorough theoretical framework for analysis, namely the dual perspective of fundamental power-status permutation and the underlying construction of meaningful order. More affirmatively, institutional arrangements for the construction of a risk culture barely occur in a unidirectional manner, as they intertwine and shape each other to form a risk society. In our study, there is a deep connection between the system of risk allocation for teachers and the subculture of professional prejudice from mass, which influence each other in ways that will be elaborated on in the last part of this article.

### 3.3. Data Collection

In this project, we seek to explore teachers' emotions and perceptions of their behavior in professional practices, including their daily classroom teaching. Among other things, teachers' perceptions of risk and their corresponding behaviors are also vital points of academic work.

The data were collected by a mixed research method. On the one hand, we mainly used a semi-structured interview guide. Interviews were conducted face-to-face. From February 2017 to December 2019, the authors interviewed 30 teachers, who were consciously chosen by variables such as age, gender, job title, and position, according to the needs of topics (e.g., 1. How do you discipline students? 2. Why do you take this approach?). In accordance with the ethical requirements of social science research, all interviewees, locations, and documents were anonymized in this study to ensure that the privacy, interests, and psychological state of local residents were not compromised. The information of interviewees is shown in Table 1. Depending on the needs of the study, two to four in-depth interviews were conducted with some of the teachers, with a total of over forty interviews, each lasting over an hour. Ultimately, our interviews resulted in 50 h of audio material, which was transcribed into text and used as our principal material. On the other hand, we conducted a questionnaire survey on emotions and perceptions among teachers across the school in 2018 (e.g., What role do you think the teacher is nowadays? What is the status relationship between teachers and students?). All teachers in L High School (N = 104) answered the

questionnaire. In addition, we went into classrooms, meeting rooms, and teacher lounges to observe teachers' emotional expressions as well as behavioral perceptions in different contexts during the survey.

**Table 1.** The basic information of interviewed teachers in L High School.

| Code | Gender | Length of Service | Subject | Academic Title | Executive Positions |
|---|---|---|---|---|---|
| TF01 | Female | 22 | English | Senior | |
| TF02 | Female | 11 | Mathematics | Junior II | |
| TF03 | Female | 3 | Biology | Junior II | |
| TF04 | Female | 5 | Mathematics | Junior II | |
| TF05 | Female | 1 | English | Junior II | |
| TF06 | Female | 1 | Chinese | Junior II | |
| TF07 | Female | 18 | English | Junior I | |
| TF08 | Female | 1 | Chinese | Junior II | |
| TF09 | Female | 22 | Mathematics | Junior I | |
| TF10 | Female | 14 | Politics | Junior I | |
| TM01 | Male | 24 | English | Senior | Deputy Head of Teaching and Research Office |
| TM02 | Male | 33 | English | Senior | Vice President |
| TM03 | Male | 26 | Chemistry | Senior | Director of Security Service |
| TM04 | Male | 3 | History | Junior II | |
| TM05 | Male | 4 | Chinese | Junior II | |
| TM06 | Male | 13 | Politics | Junior I | |
| TM07 | Male | 13 | English | Junior I | |
| TM08 | Male | 36 | Mathematics | Senior | |
| TM09 | Male | 26 | English | Senior | Deputy Head of Admissions and Communications |
| TM10 | Male | 8 | Chinese | Junior I | Deputy Secretary of the Communist Youth League |
| TM11 | Male | 18 | Chinese | Junior I | Deputy Director of Sports and Health Services |
| TM12 | Male | 23 | Mathematics | Senior | Deputy Director of Security Service |
| TM13 | Male | 15 | Geography | Junior I | |
| TM14 | Male | 27 | Biology | Senior | Head of Teaching and Research Office |
| TM15 | Male | 5 | Mathematics | Junior I | |
| TM16 | Male | 26 | History | Senior | Deputy Head of Teaching and Research Office |
| TM17 | Male | 4 | Chinese | Junior I | |
| TM18 | Male | 19 | Chinese | Senior | Head of Admissions and Communications |
| TM19 | Male | 40 | Mathematics | Senior | |
| TM20 | Male | 3 | Politics | Junior II | |

*3.4. Data Analysis*

All interviews were transcribed and analyzed inductively. The overall analytic process was an ongoing cyclical process in which categories and patterns emerged from the data and were later cross-checked [32]. During the data analysis process, NVivo 12 software was used to classify and cluster the data.

Referring to relevant research [33], some methods were adopted to ensure the trustworthiness and reliability of the data analysis. When the transcriptions were completed, they were sent back to the relative informant for cross-checking. Revisions were made when the informants had any doubts about the content of the transcription. The techniques of data triangulation and methodological triangulation were used during the process of data collection. For data triangulation, the information about teachers' emotional experiences and conception of behavior was achieved through multiple interviews with different teachers.

With regard to methodological triangulation, the data drawn from teacher interviews were compared and examined twice to attain reliable information about teachers' opinions.

Since all the teachers of L Middle School answer the questionnaire, the quantitative data are mainly analyzed by frequency analysis.

## 4. Understanding the Sense of Risk in Teachers at County High Schools

How to understand teachers' discourse of "teaching is a high-risk occupation"? William Reddy, a famous emotional historian, once put forward the concept of "emotive", pointing out that it has three functions: (1) narrative, (2) performative, and (3) emotion expressing [34]. It enlightens us that "being a teacher is a high-risk occupation" as a kind of emotive; when we examine it, we should not only see the objective facts described by the teacher but also see the emotional presentation of this discourse. Understanding teachers' sense of risk thus requires a focus on the reality of the risks faced by teachers and their subjective feelings.

### 4.1. Increased Uncertainty in Shift of Responsibility from Home Nurturing to School Education

At present, many teachers call themselves the "babysitter" of their students [35]. The term "babysitter" directly reflects the generalization of teachers' educational responsibilities. At the root of this is the shift of responsibility for family education to schools and teachers in the context of social transformation. This shift is reflected in two ways: passive shift and active shift.

On the one hand, the outflow of the labor force from rural areas has led to a passive shift in the responsibility of family education. According to our survey, 66.9% of the students at L High School are left-behind children. The temporal and spatial division caused by the exodus has made it impossible for parents to effectively take responsibility for family education. When parents are forced to hand over the guardianship of their children to the elderly, the elderly are not well placed to take on the responsibility of family education due to their limited time and knowledge, but more to take on the responsibility of living care [36,37]. As a result, the responsibility of family education is forced to shift to schools and teachers. On the other hand, some parents are also in a state of actively shifting their educational responsibilities. In particular, some parents fail to play a proper role in family education and allow their children to gradually develop uncontrollable personalities. At this point, parents will voluntarily relinquish responsibility for their children's education and expect teachers to discipline their children. As one respondent states:

> "Wisdom seems to be missing among parents in Chinese family nurturing. If a child under Grade 3 does something wrong in our schools, instead of criticizing and re-correcting him or she, parents think of their child as cute and cuddly, with a pampered mind. But as time goes on, and afterwards, the child's personality becomes more and more eccentric, and the parents themselves could not control it. At this point, the parent wants (their teacher) to make his child a stricter discipline. To put it bluntly, there are many children whose parents are unable to control them, so they want the teacher to control them". (TM17)

Whether the shift of education responsibility is passive or active, it certainly increases the responsibility of the teachers for discipline. At this time, teachers are not only responsible for teaching and educating their students but also for supervising them and keeping an eye on their safety. Correspondingly, the increased responsibility for discipline has also exposed teachers to increased uncertainty in their disciplinary practices. In particular, in the context of rapid urbanization, enrollment in county schools has gradually increased in size, resulting in progressively larger class sizes per class. At this time, county high school teachers need to manage more students. However, teachers do not have the time and energy to give adequate attention to each individual student. Especially when it comes to students' mental health, the class teacher is unable to attend to individual students due to a lack of time, energy, and even expertise.

In summary, in the context of increased educational responsibilities and rapid urbanization of education, schools and teachers face the reality of an increased element of risk, which is one of the key conditions for shaping teachers' risk perception.

*4.2. Unequal Risk Distribution and Personalized Risk Consequences*

Due to the various endowments of agents in certain fields, there are differences in their risk tolerance, although the risk is theoretically equal for all. As a result, the sequence of risk resistance differs between subjects within a risk community. "As a result of the growth and distribution of risk, certain individuals are affected more than others, that is, social risk status arises" [38].

The potential risks involved in disciplining students through a variety of distribution mechanisms ultimately make teachers the primary bearers of the risk consequences. On the one hand, society and parents are used to blaming schools and teachers for everything that concerns students. "Even if a student argues at school or falls, even if a student leaves home in a fit of rage, it is the teacher's responsibility" [39]. On the other hand, within the school system, school leaders again distribute the risks and pass them on to the teachers, as the Chinese education system stipulates that the governmental departments of the education administration can exercise management power over schools and teachers, while school administrators can manage teachers.

That is to say, schools in China have a system of headmaster responsibility and headmaster accountability. Under this dual accountability institutional arrangement, school headmasters are bound by great external control and unlimited liability [40]. At the same time, the headmaster's accountability system gives him or her greater power within the school, allowing him or her to pass on risks downwards. In practice, school administrators usually take the approach of spreading responsibility, shifting some of it to classroom teachers and regular teachers, or even putting it all on the teachers [41]. At this point, teachers have become the main risk-takers. As a teacher said: "In particular, security issues would be a vital rejected cause for a teacher's performance or promotion. When there is a safety problem, the school leader also would said that your teacher is the major responsible taker" (TM01). The *L County Teachers' Work Assessment Scale* clearly states that teachers need to "conscientiously comply with school rules and regulations, effectively fulfil the 'double responsibility' system for safety work, and actively maintain harmony and stability in the school and society".

In summary, when disciplinary risks occur, society, government administrative departments, and schools can redistribute the risk to teachers. However, teachers have to face both internal and social accountability within the education system, and they can only passively bear the adverse consequences of disciplining students.

*4.3. The Unbearability of Risk Consequences for Teachers*

In addition to the widening uncertainty, teachers also find the consequences of risk unbearable. This is particularly evident in two ways. If parents and students complain, from one side, teachers may be criticized, punished, or even fired by school leaders or education administrative departments. The emotional toll of dealing with risk in a way that, on the other side, transcends the ethics and values held by teacher holds is often unbearable.

Specifically, on the one hand, incidents of educational risk often result in excessive material, moral or professional costs for teachers. *The Law on Protection of Juveniles*, *The Law on Compulsory Education*, and *The Law on Teachers' Occupation* in China stipulate that teachers can not physically punish students; otherwise, they will be disqualified from teaching, and even then, they will never be allowed to be hired again.

For example, in 2017, the deputy principal of a secondary school in a county in Heilongjiang province was removed from his post after he quarreled with students during the process of stopping them from fighting and criticizing education (https://www.sohu.com/a/194249472_99920676, accessed 25 July 2022). Mr. L, a teacher at L High School, also ended up with RMB ¥28,000 in compensation in 2008 for a conflict with a student in

the course of disciplining him (TM03). In fact, teachers are currently often and inevitably held responsible for everything that happens to students in school, for whatever reason. It is hard even for teachers to escape responsibility for off-campus accidents involving students. In 2016, for example, when a 12-year-old student committed suicide on his way to school, the court found that two teachers were at fault and ruled that the teachers were 40% responsible for the accident (https://www.chinanews.com.cn/edu/2014/12-16/68821 83.shtml, accessed 20 August 2022). In fact, schools and teachers have become the primary responsibility for student safety accidents in and out of school. According to a survey, "schools and teachers are responsible in 90% of student injury cases, with the majority bearing primary and total responsibility" [42].

Moreover, the logic of risk allocation, which assigns responsibility to the teacher for everything that is related to students, regardless of whether they are to blame or not, has caused great harm to the teachers' own ideas of fairness and justice and has caused them great emotional damage. The survey found that the current discourse that teachers say "teaching is a high-risk profession" is also underpinned by teachers' "discontent", "insecurity", and "disappointment" with the logic of risk distribution. As stated by some respondents:

> "Most of the teachers I have come into contact with in my teaching career have always felt insecure in their teaching. For example, many students go back and jump off buildings or commit suicide because of emotional agitation, etc. Parents always feel that it is the teacher's problem or the school's problem. So we feel insecure"! (TF04)

> "I think this society, when dealing with student incidents, is not speaking up for the teacher, but everything is about safety. It does not matter whether the teacher is aggrieved or whether he or she needs to be comforted. Disappointed". (TF01)

## 5. How Teachers' Discipline Risk Is Constructed Culturally and Institutionally?

The perception of occupational risk for teachers in disciplining students in China has been articulated from two perspectives: cultural and institutional. On the one hand, the role of the teacher in traditional Chinese culture has changed dramatically under the influence of China's secularization of education, which has led to a change in cultural recognition of multiple social subjects, such as parents, schools, government, and teachers. Chinese public school, on the other hand, is a part of the governmental system, taking on some bureaucratic functions, and naturally, the teachers, as a sub-category of governmental employees, have a unique professional power, status, and mechanism of institutional functioning.

From this framework, this section examines the institutional and cultural factors that contribute to the construction of risk in the teaching profession and the mechanisms through which these two factors interact.

### 5.1. A Cultural Sight: In the Context of the Marketization and Secularization of Education, Teachers Become Service Providers with Unlimited Responsibilities

Since the time of Confucius, teaching has been held in high esteem in China. However, under China's market economy, teaching has become a commodity that can be traded in the market [43]. In some schools, particularly private schools, the relations between teachers and parents have become like those between businesses and clients [44]. In other words, the marketization and secularization of education have deconstructed the sanctity of teachers, which originally existed in traditional Chinese culture. The social members no longer see teachers as "Saints" and "Gardeners" with a cultural image full of faith and value. However, it gradually dwarfed the role of teachers as service providers. In terms of academic achievement, the role of the teacher is singularly that of a test-scoring machine for the students; in terms of school life, the teacher becomes a "good babysitter" who looks after the safety and health of the children.

The perceived cultural image of the teacher is no longer relevant to spiritual leadership, emotional nurturing, and value building. On the contrary, this cultural ground has gradually created a symbolic representation of "Customer (parent) is God" in the

educational arena. In the parents' field of vision, teachers are instrumental and functional and thus become "senior workers" with money or administrative ties. For example, the phrase is "I pay the tuition, you should provide me with all the services I need, whether you want to or not". Or, "Aren't you a teacher? You get money from the state, you are a civil servant, so you should meet all the demands of our people", etc. These public identities and symbolic discourses are turning into a culture of risk, making parents have more and more demands on teachers, and teachers have an unconditional obligation to satisfy them [45]. What, then, are the needs of parents and students? That is "babysitter" and vulgarized happy education.

Firstly, parents need teachers to be service providers for their students. Our questionnaire survey shows that 67.3% of the teachers in L middle school agree that the current "teachers are students' service providers". The "service provider" role of teachers is closely related to the expectations of some parents for their children in the context of China's social transformation. Towards the end of the math exam on 7 June 2018 (the first day of China's college entrance exam), a parent talked to another at the entrance of L High School.

> "The pressure of college entrance exams is high in S province, and basically only 1/3 of those who can be a college student. Now my child is young, and it is not possible to go out to work, so let him study at school anyway. As parents, we just need to do our best."

This educational expectation held by parents is a reflection of the reality that most students in county and rural secondary schools are "educationally hopeless" in the current distribution structure of educational opportunities [45]. This reality has also led to a shift in some parents' expectations about the professional role of teachers. For parents, if their children have a good academic foundation and are expected to go on to higher education, they will try to move to areas and schools with a higher educational level through spontaneous mobility. Many of the students who remain in county high schools have no hope of getting into university.

In the context of low expectations for their children's further education, parents' expectations for the role of teachers have changed. In the current environment, a large percentage of parents who keep their children in county high schools no longer expect teachers to be strict. Instead, parents expect teachers to take on the role of 'babysitter' and simply take on the responsibility of looking after their children at school and keeping them safe. At this point, a teacher is no longer a preacher with an exalted status but a substitute for the parents in the care of their children.

Secondly, parents need teachers to supply vulgarized, happy education. This need is closely linked to the educational philosophy that parents subscribe to. In particular, the idea of happy education, which comes from Western societies, has won the hearts of today's parents. Happy education emphasizes a higher level of pleasure based on the satisfaction of the mind and emotions rather than superficial, carnal pleasure, and therefore pleasure cannot be understood literally. However, at the same time as the idea of happy education was introduced into China and rapidly accepted, there was a tendency for society to develop the vulgarized understanding criticized by Rousseau (As Rousseau emphasizes, "vulgar theorists, who confuse indulgence with freedom, the happy child with the pampered child, must be made to understand that there is a difference.") [46]. Specifically, vulgarized happy education includes rejecting hardship, pursuing pampering, indulgence, and entertainment for all. The most obvious is the simple understanding of happy education as the pleasure of a single emotional experience. As a result, parents are more likely to try to satisfy their children's various justified or unjustified needs by coddling them during their upbringing.

There are two important reasons why parents adhere to the vulgarized idea of happy education. First, in the context of rural exodus, parents who are away from home are more likely to spoil their children with feelings of "guilty" for their absence of companionship. Since the 1980s, Chinese peasants have been moving to cities for work. However, inadequate social security systems and the low incomes of migrant workers themselves have

restricted family-based mobility and led to the accompanying phenomenon of left-behind children. In the process of mobility, the migrant parents develop a sense of guilt. This is because they are unable to care for their children after being separated from them. At this time, parents who are away from home would prefer to give their children more material compensation and happier life. It leads parents to develop an emotional state of "not wanting to scold their children". They also do not expect teachers to be harshly critical or even physically punitive in disciplining their students, but rather to grow up happily. Moreover, parents subscribe to the idea of persuasive education rather than disciplinary education. In the current social environment, they insist that they can directly guide their children well through reasoning and emotional communication rather than disciplinary education.

Second, the reality of "no hope for schooling" for county secondary school students and the insistence of parents on a "vulgarized happy education" become the theoretical underpinnings of parents who do not want teachers to take disciplinary methods such as criticism and mild corporal punishment. It also sets the stage for teachers to be held accountable by parents for problems that arise after they discipline students.

In general, as service providers, teachers are required to assume unlimited responsibility for all aspects of their students and to take the corresponding disciplinary risks. Parents, in particular, also practice the logic of "making a big fuss" over teachers' unlimited liability as a means of minimizing their own risks and securing the interests of their children. This kind of culture significantly increases the uncertainty in disciplining students' and teachers' perceptions.

*5.2. An Institutional View: The Logic of "No Accident" in Practice under a Background of "Accidental Safety Incident System"*

The "Accidental Safety Incident System" is one of the most distinctive and fundamental institutions in the Chinese education system. At numerous primary and secondary schools, it is more lethal than the college entrance rates, as this system is a powerful political and professional binding force for teachers and school leaders. It also affects a teacher's professional qualifications, title assessment, administrative promotion, financial income, and even professional honor.

In terms of content, it refers to the fact that not a single incident of malicious security, such as the death, maiming, disappearance, or fighting of a student, can occur within a school and that the teacher is the first person responsible for such incidents. In regard to approach, the "Accidental Safety Incident System" is operated through the jurisdictional authority and governmental departments to regulate schools. For example, a local Bureau of Education has a right to govern teachers by means of title assessment, occupational warning, qualification deprivation, etc. Speaking of application, it works on every ordinary teacher through various official documents and circulars. In addition, the system has developed an institutional habit, or tacit knowledge, of the practical logic of "no accident" in its daily practice. Under the practical logic of "no accident", when something happens, there is a tendency to blame schools and teachers, whether by the parents, the public, or local authorities. Moreover, when it comes to any risk accidents, the logic of the government and school is to "make things easy" or "no accident at all".

In recent years, we have often seen news of so-called "school-trouble-makers" (*xuenao*) in the online media. The production of "school-trouble-makers" is the result of the complicity of the government's logic of maintaining stability and the parents' logic of "making a big fuss". In the current environment, especially under the negative image of teachers constructed by the media and the assumption of "strong teachers, weak students", a mentality has emerged in the whole society that schools and teachers are fully responsible for what happens to students. Moreover, local authorities are supposed to uphold social justice, but in practice, they tend to act in favor of maintaining 'stability' and 'harmony' as their overriding principle.

It is crucial to note that the 'no accident' pursued by the government and schools is an extension of grassroots governance's managerial logic of no accident to the field of

education. Underlying this is the issue of school affiliation, that is, the transformation of the public school into a part of the bureaucracy [47], and thus it has a dependence on the bureaucracy. For example, L High School itself is subordinate to the L County government and is under the guidance and jurisdiction of the L County government and the Education Bureau. Based on this affiliation and dependency, the school is more accountable to the Education Bureau. The Education Bureau itself is an administrative agency, especially in the context of the emphasis on a harmonious society, where social harmony and stability are more prominently placed [48]. Taking this as the main axis, when dealing with any matter, the administrative agencies will adopt a variety of expedient governance strategies (including making peace, pacifying people, etc.) in order to smooth things over. In the case of education, if any accident happens, the administrator tends to identify it as the responsibility of the school and the teachers in the context of maintaining stability and then make the school and the teachers apologize or pay compensation in order not to have a "school crisis". The emergence of such governance strategies makes teachers feel the risks of disciplining students.

The practical logic of "No Accident" is achieved in two dimensions—formal system and informal rules. The former is a legal system of the ordinance, regulations, assessment criteria, and other explicit provisions. Students' safety and general security are explicitly listed in the performance of teachers as a part of their evaluation assessment, which is directly related to their merit pay (See *Teachers Comprehensive Evaluation Program In L High School (2011–2015)*). Meanwhile, "No Accident" has been adopted in the assessment criteria for the election of school principals. Furthermore, safety has become a key selection criterion for teacher promotion. A student's safety incident can remove a headmaster on the spot, and a teacher who had a safety incident is not eligible to be a candidate for headmaster (see *L County Primary and Secondary School Principals Performance Appraisal Trial Measures*). As a result, security has become an integral part of the career livelihood and promotion of teachers at the bottom who take greater risks. Teachers at the grassroots are more likely to be controlled by interactive techniques of excuses and evading than school leadership, so they have to take more responsibility and risks. In other words, although teachers have the least power in a safety incident, they bear most of the responsibility.

It can be seen that in the case of school risk events and logic of "No Accident" that subterfuge operates in such a way that teachers' rights and responsibilities are not equal, and their responsibilities far outweigh their powers, making them a vulnerable group.

> "Teachers are now not able to manage students in the same way as in the past. We are in a relatively weak position, and are constantly treated unfairly when conflicts arise between me and students". (TF01-1)

### 5.3. Mutual Reinforcement of Institutional and Cultural Aspects of Teachers' Discipline Risk

As noted, the constructs of discipline risk for teachers in China are both institutional and cultural. More crucially, there is also a mutual reinforcement between these two factors, which together shape teachers' high sense of risk.

In general, schools, as one of the public organizations of China, seeks to prevent potential physical, psychological, and academic risks to students at school through a rigorous control mechanism and risk allocation mechanism for teachers. However, this negative institutional mechanism of "no accident" conveys a psychological or cultural implication to parents that the school assumes all responsibility for the development of students. This transmission of a "sense of unlimited school responsibility", in turn, reinforces the sense that parents can hold the school responsible for any matters relating to their students, potentially increasing the occupational risks faced by teachers. In other words, when actors in risk societies and their governance attempt to suppress a risk using decompression mechanisms, they may create some unintended risk growth.

## 6. The Consequences of Teachers' Discipline Risks

### 6.1. Teachers' Limited Discipline Practice with a Focus on Risk Avoidance

When teachers become primary bearers of risk and perceive the consequences of these unbearable burdens, their teaching practice naturally changes dramatically. At this point, teachers' educational practice is not concerned, first and foremost, with the true goals of education but rather with how to avoid responsibility in the risk system. In other words, the teacher's personal risk aversion is a main focus of a series of educational activities. This practice of limited discipline strategies includes three typical approaches, namely, limited management, contact detachment, and "sense motive".

a.  Limited Management

In the teaching process, teachers often adopt a strategy of "limited management", which can also be called "formalistic management". It means maintaining a basic order of teaching and learning but not managing it in a higher pursuit. It has three levels of meaning. Firstly, from the perspective of the management of the whole class, it emphasizes the need to maintain some basic teaching and learning order. Secondly, it means "a limited part of the group"; teachers are only responsible for some of the students who are willing to learn. As one teacher puts it, "there are no clear rules about what teachers should do in different situations. So, teachers form a tacit agreement that 'just ignore him, and do what needs to be done and that's it. To be precise, the current teachers can say that they are responsible for those who want to learn and don't care about those who don't. That's the best option." (TM07) Thirdly, the emphasis is on separating learning from the other parts of students' lives when it comes to management, with teachers being responsible only for their learning.

b.  Contact Detachment

Contact detachment refers, on the one hand, to a reduction in contact and communication between teachers and students, and on the other hand, to the initiative of transferring problems to the class teacher rather than dealing with them directly. Specifically, the first aspect of "withdrawal" is fundamentally due to the fact that in the current environment, many subject teachers understand their role just as "teaching". Indeed, the consensus among teachers is that "doing one's job" has two dimensions: one is to avoid safety incidents in the classroom, and the other is to complete one's teaching duties, the first of which is more important than the second. On the other hand, when teachers encounter unexpected problems in the classroom, they also choose to 'pull out of contact' and pass the responsibility to the class teacher. This is because, for one thing, many teachers believe that the ordinary teacher is only responsible for the particular subject he or she is teaching and for the duration of the lesson, but not for any other problems with the students. Therefore, he only needs to be responsible for the safety of his classroom and has no responsibility to address other issues. Secondly, based on considerations of responsibility and possible risk, "there is a question of responsibility here and, to be honest, there is now a fear that something will go wrong." This is because, for teachers, there is a fear that conflict will lead to personal harm and a bad reputation. However, more critically, teachers are afraid that this will lead to unbearable burdens for themselves.

c.  Sense Motive

When teachers have to manage, they have to adopt the sense motive (*chayanguanse*) strategy. Sense motive means that teachers need to be aware of students' emotional responses at all times when disciplining them. As soon as the potential for conflict is identified, teachers will stop to discipline right now. This is because teachers are afraid that a conflict will lead to personal harm and loss of face. More importantly, however, teachers are afraid that it may lead to unacceptable risk consequences. As one teacher said, "When I see a student in a wrong state, I dare not manage again. I feel that if I manage again, he will make a fuss with me, and I can't stand it. Besides, if I hit him, I can't bear the consequences" (TF05).

*6.2. Negative Impact of Teachers' Limited Discipline on Students' Growth*

Schools are one of the key actors in the socialization of students, which plays an important role in developing their knowledge, values, and social and emotional competencies, while teachers are essential nurturers of their socialization. When teachers take risk aversion as a major consideration in their educational behavior, they inevitably fail to effectively assume the role, which in turn can have a negative impact on the socialization of students.

In terms of knowledge teaching, when teachers are only responsible for students who are willing to learn, it means that other students will fail to receive equal attention from teachers. Even more, teachers may choose to abandon unwilling students, and then the academic achievement of these students is inevitably frustrated. When it comes to social and emotional competence, as teachers adopt disciplinary strategies such as focusing only on learning and contact detachment, this means that they reduce emotional interaction with students, which also results in groups of students not receiving genuine emotional care and achievement recognition from teachers to the detriment of their emotional resilience. Considering the values dimension, when teachers are only able to "sense motive" and even cater to students in the disciplinary process, students are easily put in an awkward situation where they cannot distinguish between right and wrong, and this strategy is not conducive to the formation of students' correct values.

Generally speaking, negative disciplining practices by teachers will not be conducive to the healthy socialization of students and may even exacerbate tensions in teacher–student relationships or parents–school relationships.

## 7. Conclusions and Discussions

Drawing on the dual-theoretical perspective of risk society, this paper sheds light on the current sense of discipline risk among teachers and its roots through a mixed study in a county high school in southwest China.

The teachers' sense of discipline risk is reflected in three dimensions. First, in the context of the expansion of school's educational responsibilities and the rapid urbanization of education, county high school teachers are exposed to increased risk; second, the risk distribution mechanism in school places teachers in the role of primary risk-taker; third, the risk consequences are often unbearable for teachers. These three elements shape the objective perception of "teaching is a high-risk occupation".

Teachers' discipline risk arises at two levels. In terms of cultural construction, the secularization and marketization of education have reconstructed the teacher–student relationship and family–school relationship, turning teachers into market service providers who are unlimitedly responsive to the demands of students and parents. In county high schools, particularly, teachers are required to meet the unlimited demands of parents, including taking care of and keeping their children safe under the "pay for service" mentality, which objectively increases the risk of uncertainty for teachers. In terms of institutional construction, because the school system is embedded into the government system, the grassroots government naturally leads to a redistribution of risk within the school system in order to alter their administrative pressure on schools with a state of stay out of trouble, which ultimately makes teachers the primary bearers of discipline risk. Moreover, the two levels of risk construction reinforce each other, further reinforcing teachers' perceptions of risk. The strong sense of risk leads teachers to a state of being "afraid to discipline" or even "letting students do their own jobs", which is not conducive to the growth of students.

Therefore, it is an important issue in the current education field to eliminate teachers' experience of "high-risk occupation" and let teachers dare to discipline students. There are two ways to solve this problem. Firstly, from the perspective of defusing cultural risks, it is particularly important to reconceptualize family–school relations in line with the requirements of the era and the laws of education [49]. Education is not a market exchange relationship but rather a collaborative effort between home and school to educate people. Parents need to recognize that the payment of tuition does not mean a transfer of

responsibility for home education to school and that teachers are not saints or supermen; therefore, parents can not expect teachers to take on all the responsibilities of disciplining students. In this way, teachers and parents need to reasonably divide their duties and work closely together, reducing the uncertainty of teachers in disciplining students. In terms of resolving institutional risk, there should be a reasonable risk allocation mechanism within the education system and the school system rather than teachers being forced to be the primary bearers of risk consequences. In particular, schools should be clear about the causes of risk consequences and their division of responsibility after risk consequences have occurred [50]. The school, as an organizational body, should take responsibility for defining responsibilities fairly and for protecting teachers. In addition, the establishment of school teams consisting of educators and specialists is needed to deal with the problems of students and their families, as well as problems with discipline in the classroom. Finally, the establishment of workshops for teachers indicating how to manage the classroom and how to motivate students to learn is another solution.

In this way, teachers' discipline risk will be reduced, and it will stimulate teachers' inherent passion for being brave, bold, and willing to discipline students, turning education into "positive emotion work" once again [51].

Regarding our study, there are two main limitations. (1) The sample distribution of this study might be biased. As Table 1 shows, half of the male teachers interviewed had an executive position. Our survey found that there are no female teachers occupying administrative positions in L School. Moreover, the results of the questionnaire show that there is no significant difference in the pressure felt by teachers of different genders and teachers with or without executive positions in the aspect of "discipline students". (2) Self-reports questionnaire and interviews are used in this study, which might run the risk of socially desired responses [52]. The OECD survey of 26,000 teachers in 48 countries around the world found that the discipline of students has become a huge pressure faced by teachers around the world, which indirectly supports our research point. Therefore, we believe our survey data still have considerable credibility. As it should be, the above limitations need to be overcome in future research.

**Author Contributions:** Conceptualization, P.H. and G.T.; methodology, P.H.; formal analysis, P.H. and S.D.; investigation, P.H.; data curation, P.H.; writing-original draft preparation, P.H. and S.D.; writing-review and editing, P.H., S.D. and G.T.; project administration, P.H.; funding acquisition, P.H. All authors have read and agreed to the published version of the manuscript.

**Funding:** This research was supported by "the Fundamental Research Funds for the Central Universities" (Huazhong University of Science and Technology, 2020kfyXJJS118) and Independent Innovation Research Fund of Huazhong University of Science and Technology (2022WKYXQN008).

**Institutional Review Board Statement:** The study was conducted in accordance with the Declaration of Helsinki, and approved by the Research Ethics Committee of the School of Sociology of Huazhong University of Science and Technology in 15 September 2022.

**Informed Consent Statement:** Informed consent was obtained from all subjects involved in the study.

**Data Availability Statement:** The data are not publicly available due to confidentiality and research ethics.

**Conflicts of Interest:** The authors declare no conflict of interest. The funders had no role in the design of the study; in the collection, analyses, or interpretation of data; in the writing of the manuscript, or in the decision to publish the results.

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
