# Peer review of "Service Provider and “No Accident”: A Study of Teachers’ Discipline Risk from the Perspective of Risk Society"

_sustainability, doi:10.3390/su15054434_

Round 1
Reviewer 1 Report
Dear Editor,
Thank you for allowing me to review this manuscript. The authors utilizing a mixed design study investigated teachers' views about discipline risk from the perspective of the theory of risk society. They mainly present their findings based on the semi-structured interviews that were carried out with 30 teachers at a county town high school in Southwest China. I suggest that this work should be published but before that, some points should be fixed.
The authors should use more frequent references to support their arguments. For example, in the Conclusions and Discussions section, the references are absent.
The authors mentioned that they have used semi-structured interviews as well as observation and questionnaires. The authors have mentioned in the manuscript, findings from the questionnaire “Our questionnaire survey shows that 67.3% of the teachers in L middle school agree that the current “teachers are students’ service provider”…. Page 10”. Therefore, I see a mixed research method. Maybe the authors should revise the employed research method. Moreover, the authors should analytically present the questions from both interviews and the questionnaire.
Regarding the data analysis section, maybe, the authors used thematic analysis for their qualitative data. In this way, they should present the emerging themes. Moreover, the authors should present how did they analyze the collected data from the questionnaire.
An issue that should be discussed by the authors is the distribution of the sample. I observed that half of the male sample had an executive position. Maybe this issue led to another situation regarding the findings. The authors should give us more explanations or present this in the limitations.
Finally, in the last section, the authors should provide some limitations of their study. A common limitation is the fact that the study that uses self-reports or interviews runs the risk of socially desired responses*.
*Lavidas, K.; Papadakis, S.; Manesis, D.; Grigoriadou, A.S.; Gialamas, V. The Effects of Social Desirability on Students’ Self-Reports in Two Social Contexts: Lectures vs. Lectures and Lab Classes. Information 2022, 13, 491. https://doi.org/10.3390/info13100491
Author Response
Response to Reviewer 1
Dear reviewer,
We appreciate your generous suggestions on this article which are crucial to a further development of our research. We are glad to accept them and have made the following revisions.
- The authors should use more frequent references to support their arguments. For example, in the Conclusions and Discussions section, the references are absent.
Response: Thank you very much. We have added 7 latest references to support their arguments included the section of Conclusions and Discussions. Please see them in page 15-16.
- Maybe the authors should revise the employed research method. Moreover, the authors should analytically present the questions from both interviews and the questionnaire. Moreover, the authors should present how did they analyze the collected data from the questionnaire.
Response: Thank you very much. we have revise our research method (including data collection and data analysis) in page 6-7.
- An issue that should be discussed by the authors is the distribution of the sample. I observed that half of the male sample had an executive position. Maybe this issue led to another situation regarding the findings. The authors should give us more explanations or present this in the limitations.
Response: Thank you very much. We have improved the discussion on our distribution of the sample in page 16
- Finally, in the last section, the authors should provide some limitations of their study. A common limitation is the fact that the study that uses self-reports or interviews runs the risk of socially desired responses.
Response: Thank you very much. We have added some limitations of this study in the last paragraphs in page 16.
Please see the revised version of our article for details. We look forward to your further feedback and suggestions. Thank you so much!
Sincerely,
Penghui Hu
Shasha Du
Guoxiu Tian
Reviewer 2 Report
|
Section |
Comments |
|
Title |
Service provider and “No Accidents”: A qualitative study of teachers’ discipline risk from the perspective of risk society |
|
Overall |
Study describes risk of teachers in China in the tension between providing education and discipline for students in balance with societal expectations and limitations (home environment of students in particular).
Well written paper – some grammatical issues throughout that will be addressed with a thorough read-through.
Important, timely topic and well supported with literature and data.
|
|
Abstract |
Abstract accurately states the conclusion of the study.
|
|
Introduction |
Solid lit review that sets the stage for the current study. Thorough overview of risk theory, cultural expectations for teachers in China, and the confounding factors leading to educator risk.
|
|
Materials and Methods |
Clear and logical explanation of methods. Methods align with research questions and study conclusions.
Providing sample questions from the semi-structured interview would be helpful to the reader.
|
|
Results |
Adding a ‘results’ heading and additional signposts throughout the results/discussion section would help orient the reader. There is a lot of good information included in the discussion, but easy to get lost. Consider improving the overall structure and perhaps moving the numbered items out as subsection headers.
Consider adding a study limitations section.
|
|
Conclusions |
Conclusions align with research questions and results |
Author Response
Response to Reviewer 2
Dear reviewer,
We appreciate your generous suggestions on this article which are crucial to a further development of our research. We are glad to accept them and have made the following revisions.
- Providing sample questions from the semi-structured interview would be helpful to the reader.
Response: We have added 4 examples of question from our semi-structured interview at 3.3 Date Collection in Page 6.
2.Consider adding a study limitations section.
Response: We have added a part of limitations in the last of our paper in page 16.
- Adding a ‘results’ heading and additional signposts throughout the results/discussion section would help orient the reader.
Response: We have added the ‘results’ heading in page 7, and added other signposts in corresponding sections.
Please see the revised version of our article for details. We look forward to your further feedback and suggestions. Thank you so much!
Sincerely,
Penghui Hu
Shasha Du
Reviewer 3 Report
The manuscript aims to explore the institutional structure, cultural background, and mechanisms behind the experience of county town Chinese teachers' sense of risk in disciplining students, as well as its consequences. The beginning of the title (Service provider and "No Accidents") only becomes clear to the reader in the second part of the text. It may be worth (for greater clarity ) abandoning it. The manuscript is presented in a well-structured manner. The presentation of the problem, the research and the discussion form a logical sequence, leading to important recommendations for the Chinese system of education. Here are some detailed suggestions:
1.Introduction
An important problem present in the experience of teachers, yet underrepresented in research, has been pointed out. The second research problem should be made more specific
2.Literature Review
In discussing the phenomenon under analysis, the authors have followed a logical sequence. The point of reference is the theories of the risk society. I have marked the less readable passages in the text in yellow.
2.1 Risk society: the approaches of institutionalism and culturism
The language is not always clear (text marked in yellow). I suggest consistently giving or not giving the names of the authors of the works cited. It is advisable to clarify what is characteristic of early modernity and what is characteristic of late modernity and what change in the normative framework is associated with this transition. The sentence "The creation of a range of institutions in recent times has provided the context for the realization of these two contradictory orientations as well as the normative framework" therefore needs to be clarified.
I suggest mentioning the names of social scientists of culturism in the text. It is not clear what programs the authors are referring to in the sentence "Fenwick observes the emergence of specific norms of technological internalization and self-regulation that "good" teachers deploy themselves through such programs." Similarly, in the sentence "A report explores..." it is useful to specify at the beginning of the sentence the report where and when it was conducted.
3 Methodology
The specifics of the study area are described. Suggests moving 3.2. Theory to the theory section. Data collection and analysis was clearly presented. To ensure reliability of the data analysis data triangulation and methodological triangulation techniques were used.
For better readability of the text, I suggest that before quoting the respondents' statements, I would add: As one respondent states.... In 2.Unequal risk distribution and personalized risk consequences I suggest illustrating the considerations with a quote from the respondents' statements.
2.The unbearability of risk consequences for teachers: it is not entirely clear what vulgarization in happy education consists of. It is worth clarifying whether China has a Convention on the Rights of the Child prohibiting physical punishment.
Discussion
I suggest adding to the solutions the establishment of school teams consisting of educators and specialists to deal with the problems of students, their families, as well as problems with discipline in the classroom, lack of motivation to learn. Another solution is workshops for teachers indicating how to manage the classroom, how to motivate students to learn.
The bibliography list is comprehensive, although 31 items of bibliography published before 2017.

Author Response
Response to Reviewer 3
Dear reviewer,
We appreciate your generous suggestions on this article which are crucial to a further development of our research. We are glad to accept them and have made the following revisions.
- The beginning of the title (Service provider and "No Accidents") only becomes clear to the reader in the second part of the text. It may be worth (for greater clarity ) abandoning it.
Response: Thank you very much. In our opinion, the title (Service provider and "No Accidents") is our answer to why teachers face high degree of risk when disciplining students, and it is also the key point of this paper. So, we think that maybe it is worth remaining.
- An important problem present in the experience of teachers, yet underrepresented in research, has been pointed out. The second research problem should be made more specific.?
Response: Thank you very much. We have revised the expression of the second research question to make it more specific in page 2.
- In discussing the phenomenon under analysis, the authors have followed a logical sequence. The point of reference is the theories of the risk society. I have marked the less readable passages in the text in yellow.
Response: Thank you very much. We have revised the less readable passages you marked in yellow. Please see them in page 2, 3, 4, 5
- I suggest mentioning the names of social scientists of culturism in the text .It is not clear what programs the authors are referring to in the sentence "Fenwick observes the emergence of specific norms of technological internalization and self-regulation that "good" teachers deploy themselves through such programs." Similarly, in the sentence "A report explores..." it is useful to specify at the beginning of the sentence the report where and when it was conducted.
Response: Thank you very much. We have added the names of culturism scholars in page 3 as well as some details of the report
- The specifics of the study area are described. Suggests moving 3.2. Theory to the theory section.
Response: Thanks very much for your suggestion. We have changed “3.2 Theory” to “3.2 Theoretical Frame”, and we are of the opinion that this part plays the role as introduction of how we use the theory of risk society to analyze teachers’ discipline risk. And, we think it is suitable to put it in this place.
- For better readability of the text, I suggest that before quoting the respondents' statements, I would add: As one respondent states.... In 2.Unequal risk distribution and personalized risk consequences I suggest illustrating the considerations with a quote from the respondents' statements.
Response: Thank you very much .We have improved this point with whole paper, such as page 9.
- The unbearability of risk consequences for teachers: it is not entirely clear what vulgarization in happy education consists of. It is worth clarifying whether China has a Convention on the Rights of the Child prohibiting physical punishment.
Response: Thank you very much. We have improved “what vulgarization in happy education consists of” in page 11. And we have added the related statement of “Child prohibiting physical punishment” in page 9.
- I suggest adding to the solutions the establishment of school teams consisting of educators and specialists to deal with the problems of students, their families, as well as problems with discipline in the classroom, lack of motivation to learn. Another solution is workshops for teachers indicating how to manage the classroom, how to motivate students to learn.
Response: Thanks for your suggestion very much. We have added them as the solution to solve teachers’ sense of risk. Please see page15.
Please see the revised version of our article for details. We look forward to your further feedback and suggestions. Thank you so much!
Sincerely,
Penghui Hu
Shasha Du
Guoxiu Tian